# STEP SIZE OPTIMIZATION

## ABSTRACT

This paper proposes a new approach for step size adaptation in gradient methods. The proposed method called step size optimization (SSO) formulates the step size adaptation as an optimization problem which minimizes the loss function with respect to the step size for the given model parameters and gradients. Then, the step size is optimized based on alternating direction method of multipliers (ADMM). SSO does not require the second-order information or any probabilistic models for adapting the step size, so it is efficient and easy to implement. Furthermore, we also introduce stochastic SSO for stochastic learning environments. In the experiments, we integrated SSO to vanilla SGD and Adam, and they outperformed state-of-the-art adaptive gradient methods including RMSProp, Adam, $L^4$-Adam, and AdaBound on extensive benchmark datasets.

## 1 INTRODUCTION

First-order gradient methods (simply gradient methods) have been widely used to fit model parameters in machine learning and data mining, such as training deep neural networks. In the gradient methods, step size (or learning rate) is one of the most important hyperparameters that determines the overall optimization performance. For this reason, step size adaptation has been extensively studied from various perspectives such as second-order information (Byrd et al., 2016; Schaul et al., 2013), Bayesian approach (Mahsereci & Henning, 2015), learning to learn paradigm (Andrychowicz et al., 2016), and reinforcement learning (Li & Malik, 2017). However, they are hardly used in practice due to lack of solid empirical evidence for the step size adaptation performance, hard implementation, or huge computation. For these reasons, some heuristically-motivated methods such as AdaGrad (Duchi et al., 2011), RMSProp (Tieleman & Hinton, 2012), and Adam (Kingma & Ba, 2015) are mainly used in practice to solve the large-scale optimization problems such as training deep neural networks.

Recently, two impressive methods, called $L^4$ (Rolinek & Martius, 2018) and AdaBdound (Luo et al., 2019), were proposed to efficiently adapt the step size in training of models, and showed some improvement over existing methods without huge computation. However, performance comparisons to them were conducted only on relatively simple datasets such as MNIST and CIFAR-10, even though $L^4$ has several newly-introduced hyperparameters, and AdaBound needs manually-desgined bound functions. Moreover, $L^4$ still requires about 30% more execution time, and AdaBound lacks the time complexity analysis or empirical results on training performance against actual execution time.

This paper proposes a new optimization-based approach for the step size adaptation, called step size optimization (SSO). In SSO, the step size adaptation is formulated as a sub-optimization problem of the gradient methods. Specifically, the step size is adapted to minimize a linearized loss function for the current model parameter values and gradient. The motivation of SSO and the justification for the performance improvement by SSO is clear because it directly optimizes the step size to minimize the loss function. We also present a simple and efficient algorithm to solve this step size optimization problem based on the alternating direction method of multipliers (ADMM) (Gabay & Mercier, 1976). Furthermore, we provide a practical implementation of SSO on the loss function with $L_2$ regularization (Krogh & Hertz, 1992) and stochastic SSO for the stochastic learning environments.

SSO does not require the second-order information (Byrd et al., 2016; Schaul et al., 2013) and any probabilistic models (Mahsereci & Henning, 2015) to adapt the step size, so it is efficient and easy to implement. We analytically and empirically show that the additional time complexity of SSO in

the gradient methods is negligible in the training of the model. To validate the practical usefulness of SSO, we made two gradient methods, SSO-SGD and SSO-Adam, by integrating SSO to vanilla SGD and Adam. In the experiments, we compared the training performance of SSO-SGD and SSO-Adam with two state-of-the-art step size adaptation methods ($L^4$ and AdaBdound) as well as the most commonly used gradient methods (RMSProp and Adam) on extensive benchmark datasets.

## 2 STEP SIZE OPTIMIZATION

### 2.1 PROBLEM FORMULATION

The goal of step size optimization (SSO) is to find the optimal step size that minimizes the loss function with respect to the step size $\eta$ as:

$$\eta^* = \arg\min_{\eta} \mathcal{J}(\boldsymbol{\theta} - \eta\mathbf{v}) + \Omega(\boldsymbol{\theta} - \eta\mathbf{v}), \tag{1}$$

where $\mathcal{J}$ is the loss function; $\Omega$ is a regularization term; $\boldsymbol{\theta}$ is the model parameter; and $\mathbf{v}$ is the gradient for updating $\boldsymbol{\theta}$. Note that $\mathbf{v}$ is an optimizer-dependent gradient such as the moving average of the gradients in Adam. As gradient methods update the model by moving to the opposite direction of the gradient ($\boldsymbol{\theta} \leftarrow \boldsymbol{\theta} - \eta\mathbf{v}$), the loss function $\mathcal{J}(\boldsymbol{\theta})$ and the regularization term $\Omega(\boldsymbol{\theta})$ can be expressed as $\mathcal{J}(\boldsymbol{\theta} - \eta\mathbf{v})$ and $\Omega(\boldsymbol{\theta} - \eta\mathbf{v})$, respectively. In real-world problems, however, directly solving the optimization problem in Eq. (1) is infeasible due to the severe nonlinearity of $\mathcal{J}$. To handle this difficulty, first, we linearize $\mathcal{J}$ around $\boldsymbol{\theta}$ as:

$$\mathcal{J}(\boldsymbol{\theta} - \eta\mathbf{v}) \approx \mathcal{J}(\boldsymbol{\theta}) + (\nabla_{\boldsymbol{\theta}}\mathcal{J})^T(\boldsymbol{\theta} - \eta\mathbf{v} - \boldsymbol{\theta}) = \mathcal{J}(\boldsymbol{\theta}) - \eta\mathbf{g}^T\mathbf{v}, \tag{2}$$

where $\mathbf{g} = \nabla_{\boldsymbol{\theta}}\mathcal{J}$ is the true gradient. Note that $\mathbf{v}$ is the same as $\mathbf{g}$ in vanilla gradient method.

However, in order for the linearization of Eq. (2) to be valid, $\eta$ should be sufficiently small. To this end, we introduce an inequality constraint for the upper bound of $\eta$. Thus, the optimization problem of SSO is given by:

$$\eta^* = \arg\min_{0 \le \eta \le \epsilon} \mathcal{J}(\boldsymbol{\theta}) - \eta\mathbf{g}^T\mathbf{v} + \Omega(\boldsymbol{\theta} - \eta\mathbf{v}), \tag{3}$$

where $\epsilon$ is a positive hyperparameter that defines the upper bound of the step size. That is, SSO adapts the step size by solving the constrained optimization problem in Eq. (3).

### 2.2 AUGMENTED LAGRANGIAN FOR STEP SIZE OPTIMIZATION PROBLEM

The augmented Lagrangian is a widely used optimization technique to handle a constrained optimization problem by transforming it into an unconstrained problem. The objective function in the augmented Lagrangian is defined based on equality constraints. For this reason, we need to transform the optimization problem with the inequality constraints in Eq. (3) to the problem with the equality constraints by introducing slack variables $s_1$ and $s_2$ as:

$$\eta^* = \arg\min_{\eta} \mathcal{J}(\boldsymbol{\theta}) - \eta\mathbf{g}^T\mathbf{v} + \Omega(\boldsymbol{\theta} - \eta\mathbf{v}) \tag{4}$$
$$\text{s. t. } \eta - s_1 = 0, \epsilon - \eta - s_2 = 0$$
$$s_1 \ge 0, s_2 \ge 0.$$

Finally, the augmented Lagrangian for the problem of Eq. (4) is given by:

$$\mathcal{L}_{\mu}(\eta, \lambda_1, \lambda_2, s_1, s_2) = \mathcal{J}(\boldsymbol{\theta}) - \eta\mathbf{g}^T\mathbf{v} + \Omega(\boldsymbol{\theta} - \eta\mathbf{v}) - \lambda_1(\eta - s_1) - \lambda_2(\epsilon - \eta - s_2)$$
$$+ \frac{\mu}{2}(\eta - s_1)^2 + \frac{\mu}{2}(\epsilon - \eta - s_2)^2, \tag{5}$$

where $\lambda_1 \ge 0$ and $\lambda_2 \ge 0$ are dual variables, and $\mu$ is a balancing parameter between the objective function and the penalty term for the equality constraints. In general, $\mu$ is simply set to be gradually increased in optimization process to guarantee the feasibility for the equality constraints (Gabay & Mercier, 1976; Ouyan et al., 2013).

### 2.3 OPTIMIZATION

In this section, we describe the optimization algorithm to find the optimal step size that minimizes the augmented Lagrangian in Eq. (5) using ADMM. In mathematical optimization and machine learning, ADMM has been widely used to solve the optimization problem containing different types of primal variables $\mathbf{x}$, $\mathbf{z}$ with the equality constraints such as:

$$\mathbf{x}, \mathbf{z} = \arg\min_{\mathbf{x}, \mathbf{z}} f(\mathbf{x}) + g(\mathbf{z}) \tag{6}$$

$$\text{s. t. } \boldsymbol{A}\mathbf{x} + \boldsymbol{B}\mathbf{z} = \mathbf{c},$$

where $\boldsymbol{A}$ and $\boldsymbol{B}$ are coefficient matrices of the equality constraints, and $\mathbf{c}$ is a constant. ADMM iteratively finds the optimal variables by minimizing the augmented Lagrangian for the problem in Eq. (6), denoted by $\mathcal{L}_\mu(\mathbf{x}, \mathbf{z}, \boldsymbol{\lambda})$, because directly solving the problem can be nontrivial. Specifically, ADMM (Algorithm 1) optimizes primal and dual variables in a one-sweep Gauss-Seidel manner: $\mathcal{L}_\mu(\mathbf{x}, \mathbf{z}, \boldsymbol{\lambda})$ is minimized with respect to the primal variables $\mathbf{x}$ and $\mathbf{z}$ alternatively for the fixed dual variable $\boldsymbol{\lambda}$. Then, $\mathcal{L}_\mu(\mathbf{x}, \mathbf{z}, \boldsymbol{\lambda})$ is minimized over the dual variable $\boldsymbol{\lambda}$ for the fixed primal variables $\mathbf{x}$ and $\mathbf{z}$.

---

**Algorithm 1:** ADMM

**Output:** Optimized primal variables $\mathbf{x}$ and $\mathbf{z}$

1   $t = 0, \boldsymbol{\lambda} = 0, \mu = 1$
2   **repeat**
3      $\mathbf{x}^{(t+1)} \leftarrow \arg\min_{\mathbf{x}} \mathcal{L}_\mu(\mathbf{x}, \mathbf{z}^{(t)}, \boldsymbol{\lambda}^{(t)})$
4      $\mathbf{z}^{(t+1)} \leftarrow \arg\min_{\mathbf{z}} \mathcal{L}_\mu(\mathbf{x}^{(t+1)}, \mathbf{z}, \boldsymbol{\lambda}^{(t)})$
5      $\boldsymbol{\lambda}^{(t+1)} \leftarrow \boldsymbol{\lambda}^{(t)} - \mu(\boldsymbol{A}\mathbf{x}^{(t+1)} + \boldsymbol{B}\mathbf{z}^{(t+1)} - \mathbf{c})$
6      Increase $\mu$
7      $t \leftarrow t + 1$
8   **until** $\mathbf{x}, \mathbf{z}, \boldsymbol{\lambda}$ converged;

---

The step size optimization problem of Eq. (4) is an equality constrained problem, and also has two kinds of primal variables, the step size $\eta$ and the slack variables $s_1, s_2$. That is, for the primal variables $\eta$ and $s_1, s_2$, the step size optimization problem of Eq. (4) has the same structure as the problem in Eq. (6), where $f(\eta) = \mathcal{J}(\boldsymbol{\theta}) - \eta \mathbf{g}^T \mathbf{v} + \Omega(\boldsymbol{\theta} - \eta \mathbf{v})$, $g(s_1, s_2) = 0$, and the equality constraints are $\eta - s_1 = 0$ and $\epsilon - \eta - s_2 = 0$. Thus, the primal variables of the step size optimization problem $\eta, s_1, s_2$, and the dual variables $\lambda_1, \lambda_2$ can be optimized by ADMM as:

$$\eta^{(t+1)} \leftarrow \arg\min_\eta \mathcal{L}_\mu(\eta, s_1^{(t)}, s_2^{(t)}, \lambda_1^{(t)}, \lambda_2^{(t)}), \tag{7}$$

$$s_1^{(t+1)} \leftarrow max\left\{0, \arg\min_{s_1} \mathcal{L}_\mu(\eta^{(t+1)}, s_1, s_2^{(t)}, \lambda_1^{(t)}, \lambda_2^{(t)})\right\}, \tag{8}$$

$$s_2^{(t+1)} \leftarrow max\left\{0, \arg\min_{s_2} \mathcal{L}_\mu(\eta^{(t+1)}, s_1^{(t+1)}, s_2, \lambda_1^{(t)}, \lambda_2^{(t)})\right\}, \tag{9}$$

$$\lambda_1^{(t+1)} = max\left\{0, \lambda_1^{(t)} - \mu(\eta^{(t+1)} - s_1^{(t+1)})\right\}, \tag{10}$$

$$\lambda_2^{(t+1)} = max\left\{0, \lambda_2^{(t)} - \mu(\epsilon - \eta^{(t+1)} - s_2^{(t+1)})\right\}. \tag{11}$$

Note that the $max$ operation with zero is applied to Eq. (8) $\sim$ (11) for satisfying the nonnegative constrains of the slack and the dual variables.

Algorithm 2 shows the overall process of the gradient descent method with SSO to optimize $\boldsymbol{\theta}$. In line 3, the true gradient $\mathbf{g}$ and the adaptive gradient $\mathbf{v}$ are computed, and the augmented Lagrangian $\mathcal{L}_\mu(\eta, s_1, s_2, \lambda_1, \lambda_2)$ for optimizing the step size is determined based on the current gradients. Then, the step size is optimized for currently given model parameters and gradients by iteratively minimizing $\mathcal{L}_\mu(\eta, s_1, s_2, \lambda_1, \lambda_2)$ in line 5~13. Finally, $\boldsymbol{\theta}$ is updated with the optimized step size and the adaptive gradient in line 14.

---

**Algorithm 2:** Gradient descent method with SSO

---

**Input** : Upper bound of the step size: $\epsilon \in (0, 1]$
**Output:** $\theta$

1   $k = 0$
2   **repeat**
3      Compute $\mathbf{g} = \nabla_\theta \mathcal{J}$ and $\mathbf{v}$ to determine $\mathcal{L}_\mu(\eta, s_1, s_2, \lambda_1, \lambda_2)$
4      $t = 0, s_1 = 0, s_2 = 0, \lambda_1 = 0, \lambda_2 = 0, \mu = 1$
5      **repeat**
6         $\eta^{(t+1)} \leftarrow \arg\min_\eta \mathcal{L}_\mu(\eta, s_1^{(t)}, s_2^{(t)}, \lambda_1^{(t)}, \lambda_2^{(t)})$
7         $s_1^{(t+1)} \leftarrow max \left\{ 0, \ \arg\min_{s_1} \mathcal{L}_\mu(\eta^{(t+1)}, s_1, s_2^{(t)}, \lambda_1^{(t)}, \lambda_2^{(t)}) \right\}$
8         $s_2^{(t+1)} \leftarrow max \left\{ 0, \ \arg\min_{s_2} \mathcal{L}_\mu(\eta^{(t+1)}, s_1^{(t+1)}, s_2, \lambda_1^{(t)}, \lambda_2^{(t)}) \right\}$
9         $\lambda_1^{(t+1)} \leftarrow max \left\{ 0, \ \lambda_1^{(t)} - \mu(\eta^{(t+1)} - s_1^{(t+1)}) \right\}$
10        $\lambda_2^{(t+1)} \leftarrow max \left\{ 0, \ \lambda_2^{(t)} - \mu(\epsilon - \eta^{(t+1)} - s_2^{(t+1)}) \right\}$
11        Increment $\mu$
12        $t \leftarrow t + 1$
13      **until** $\eta$ converged;
14      $\boldsymbol{\theta}^{(k+1)} \leftarrow \boldsymbol{\theta}^{(k)} - \eta\mathbf{v}$
15      $k \leftarrow k + 1$
16 **until** $\boldsymbol{\theta}$ converged;

---

## 2.4   UPPER BOUND DECAY

In this section, we additionally introduce the upper bound decay for SSO. Since the shape of the augmented Lagrangian for SSO can be changed by the regularization term $\Omega$, the convergence property of the gradient methods with SSO may be different for each different regularization term. To overcome this problem, we devise the upper bound decay method and integrate it into SSO. One possible implementation of the upper bound decay is to use the exponential decay as follows.

$$\epsilon^{(k+1)} \leftarrow \gamma\epsilon^{(k)}, \tag{12}$$

where $\gamma \in (0, 1)$ is decay factor. That is, $\epsilon$ is exponentially decreased over the training.

It is similar to the step size decay, but there is a big difference. The upper bound decay indirectly reduces the step size by decreasing the upper bound of the step size instead of reducing it directly. That is, SSO with the upper bound decay automatically provides an optimal step size that will be gradually reduced over the training. Furthermore, SSO with the upper bound decay always guarantees that the step size converges to zero regardless of the shape of the augmented Lagrangian for the valid decay factors such as $\gamma \in (0, 1)$ in Eq (12). Thus, the upper bound decay is more flexible than the step size decay and can be regarded as a generalized method of the step size decay. One main advantage of SSO over the existing methods is that it can exploit such upper bound decay. In SSO with the upper bound decay, the initial upper bound $\epsilon^{(0)}$ is a hyperparameter.

## 2.5   PRACTICAL IMPLEMENTATION: SSO WITH $L_2$ REGULARIZATION

In this section, we derive SSO with the $L_2$ regularization that is the most widely used regularization technique and also provide a practical implementation of the gradient method with SSO.

### 2.5.1   UPDATE RULE

With the $L_2$ regularization term, the augmented Lagrangian of the step size optimization problem is given by:

$$\mathcal{L}_\mu(\eta, \lambda_1, \lambda_2, s_1, s_2) = \mathcal{J}(\boldsymbol{\theta}) - \eta\mathbf{g}^T\mathbf{v} + \frac{\beta}{2}||\boldsymbol{\theta} - \eta\mathbf{v}||_2^2 - \lambda_1(\eta - s_1) - \lambda_2(\epsilon - \eta - s_2)$$
$$+ \frac{\mu}{2}(\eta - s_1)^2 + \frac{\mu}{2}(\epsilon - \eta - s_2)^2, \tag{13}$$

where $\beta$ is a positive hyperparameter of the $L_2$ regularization for balancing the loss function and the regularization term. By applying ADMM in Algorithm 1, we can optimize the step size using the following update rules:

$$\eta^{(t+1)} = \frac{\mathbf{g}^T\mathbf{v} + \beta\mathbf{v}^T\boldsymbol{\theta} + \lambda_1^{(t)} - \lambda_2^{(t)} + \mu(s_1^{(t)} - s_2^{(t)} + \epsilon)}{\beta\mathbf{v}^T\mathbf{v} + 2\mu} \tag{14}$$

$$s_1^{(t+1)} = max\left\{0,\ \eta^{(t+1)} - \frac{\lambda_1^{(t)}}{\mu}\right\} \tag{15}$$

$$s_2^{(t+1)} = max\left\{0,\ \epsilon - \eta^{(t+1)} - \frac{\lambda_2^{(t)}}{\mu}\right\}. \tag{16}$$

Note that the update rule for the dual variables $\lambda_1$ and $\lambda_2$ are the same as Eq. (10) and (11) because the update rules of the dual variables are independent of the loss function and the regularization term in SSO. More precisely, the update rules of the dual variables depend only on the equality constraints for the step size.

### 2.5.2 CONVERGENCE

If the optimal step size exists within an range $[0, \epsilon)$, the slack variables converge as $s_1 \to \eta$ and $s_2 \to \epsilon - \eta$ when ADMM in SSO is sufficiently iterated ($\mu \to \infty$). Thus, the step size $\eta$ converges to some value as:

$$\eta^{(t+1)} \to \frac{\mathbf{g}^T\mathbf{v} + \beta\mathbf{v}^T\boldsymbol{\theta} + \lambda_1^{(t)} - \lambda_2^{(t)} + 2\mu\eta^{(t)}}{\beta\mathbf{v}^T\mathbf{v} + 2\mu} \approx \frac{2\mu\eta^{(t)}}{2\mu} = \eta^{(t)}. \tag{17}$$

In contrast, if the optimal step size exists over the upper bound, the step size may converge near the upper bound ($\eta \approx \epsilon$) in ADMM. Thus, the slack variables converge as $s_1 \to \epsilon$ and $s_2 \to 0$, and the step size consequently converge to the upper bound as follows.

$$\eta^{(t+1)} \to \frac{\mathbf{g}^T\mathbf{v} + \beta\mathbf{v}^T\boldsymbol{\theta} + \lambda_1^{(t)} - \lambda_2^{(t)} + 2\mu\epsilon}{\beta\mathbf{v}^T\mathbf{v} + 2\mu} \approx \frac{2\mu\epsilon}{2\mu} = \epsilon. \tag{18}$$

Thus, the step size always converges in ADMM of SSO.

Unfortunately, the second case ($\eta \to \epsilon$) is not the desired result in which the model is sufficiently trained because a relatively small step size is required in this situation to make the gradient methods converge. However, it is not a problem in SSO with the upper bound decay because the upper bound $\epsilon$ must be reduced by the decay method over the training.

## 3 STOCHASTIC SSO

In this section, we describe SSO for the stochastic learning environments and also provide stochastic SSO with $L_2$ regularization. The optimization problem in the stochastic environments can be formulated on a mini-batch with respect to the step size as follows.

$$\eta^{(k+1)} = \underset{0\leq\eta\leq\epsilon}{\arg\min} \frac{1}{N}\sum_{i=1}^{N}\mathcal{J}_i(\boldsymbol{\theta}^{(k)} - \eta\mathbf{v}^{(k)}) + \Omega(\boldsymbol{\theta}^{(k)} - \eta\mathbf{v}^{(k)}), \tag{19}$$

where $\mathcal{J}_i$ is the loss function for the $i^{\text{th}}$ sample in the mini-batch. This problem can be rewritten as an optimization problem with a conservative penalty term as (Ouyan et al., 2013; Li et al., 2014):

$$\eta^{(k+1)} = \underset{0\leq\eta\leq\epsilon}{\arg\min} \frac{1}{N}\sum_{i=1}^{N}\mathcal{J}_i(\boldsymbol{\theta}^{(k)} - \eta\mathbf{v}^{(k)}) + \Omega(\boldsymbol{\theta}^{(k)} - \eta\mathbf{v}^{(k)}) + \frac{||\boldsymbol{\theta}^{(k)} - \boldsymbol{\theta}^{(k+1)}||_2^2}{\eta}. \tag{20}$$

Note that the conservative penalty term is introduced to prevent undesired large change in the model parameters as mini-batch changes. The conservative penalty term can be rewritten with respect to the step size as:

$$\frac{||\boldsymbol{\theta}^{(k)} - \boldsymbol{\theta}^{(k+1)}||_2^2}{\eta} = \frac{||\boldsymbol{\theta}^{(k)} - (\boldsymbol{\theta}^{(k)} - \eta\mathbf{v}^{(k)})||_2^2}{\eta} = \eta\mathbf{v}^{(k)^T}\mathbf{v}^{(k)}. \tag{21}$$

Thus, we can derive the optimization algorithm for $\eta^{(k+1)}$ in stochastic environments by applying the linearization and ADMM to the problem in Eq. (20) as described in Section 2.1 $\sim$ 2.3.

In addition, we provide the update rule of stochastic SSO with $L_2$ regularization for the practical implementation of it. Similar to the update rule of the deterministic SSO with $L_2$ regularization in Section 2.5.1, the update rule for $\eta$ of stochastic SSO with $L_2$ regularization is given by:

$$\eta^{(k+1)} = \frac{\frac{1}{N}\sum_{i=1}^{N}\mathbf{g}_i^{(k)^T}\mathbf{v}^{(k)} - \mathbf{v}^{(k)^T}\mathbf{v}^{(k)} + \beta\mathbf{v}^{(k)^T}\boldsymbol{\theta}^{(k)} + \lambda_1^{(t)} - \lambda_2^{(t)} + \mu(s_1^{(t)} - s_2^{(t)} + \epsilon)}{\beta\mathbf{v}^{(k)^T}\mathbf{v}^{(k)} + 2\mu} \quad (22)$$

Note that the update rule of the slack variables are the same as Eq. (15) and (16) because they depend only on the constraints of the problem.

## 4   Time Complexity Analysis

Due to the sub-optimization process for the step size adaptation of SSO, a gradient method with SSO inevitably requires additional complexity. In this section, we analyze the time complexity of the gradient method with SSO and show that the additional time complexity of the gradient method with SSO is almost the same as the vanilla gradient method in large-scale optimization problems such as training deep neural networks. The empirical time complexity analysis over the actual execution time will be conducted in the experiment section.

The time complexity of a gradient method with SSO is $O(T\psi + Td + TI)$, where $T$ is the number of epochs in the gradient method; $\psi$ is the total computation for model updates such as the forward and the backward steps in neural networks; $d$ is the number of model parameters; and $I$ is the number of iterations for updating $\eta$ (line 5$\sim$13 in Algorithm 2). The additional time complexity from SSO is $O(Td + TI)$. The time complexity $O(Td)$ comes from the dot product operations to compute $\mathbf{g}^T\mathbf{v}, \mathbf{v}^T\boldsymbol{\theta}$, and $\mathbf{v}^T\mathbf{v}$. Another time complexity $O(TI)$ comes from the iterative optimization process of ADMM for adapting $\eta$. Practically, however, $O(T\psi + Td + TI) \approx O(T\psi)$ because the dot product operation can be accelerated significantly by GPU, and $O(Td + TI)$ is negligible compared to $O(T\psi)$ when we use a sophisticated model for high representation ability, such as a deep neural network. Thus, in real-world applications, the time complexity of the gradient method with SSO is approximately the same as $O(T\psi)$, which is the time complexity of the vanilla gradient method. In the experiment section, this time complexity analysis for SSO will be also demonstrated empirically on several real-world datasets.

## 5   Experiments

In the experiments, we validated the effectiveness of SSO in gradient-based optimization. To this end, we generated two gradient methods with stochastic SSO using the upper bound decay – (1) vanilla SGD with SSO (SSO-SGD) and (2) Adam with SSO (SSO-Adam), and then their optimization performance was compared with the state-of-the-art gradient methods: 1) RMSProp; 2) Adam; 3) $L^4$-Adam; 4) AdaBound. We specified the optimization problem as training deep neural networks because it is the most appealing and challenging problem using gradient methods. To train deep neural networks, cross-entropy loss with $L_2$ regularization is used as a loss function.

We conducted the experiments on four well-known benchmark datasets: MNIST, SVHN, CIFAR-10, and CIFAR-100 datasets. For MNIST dataset, convolutional neural network (CNN) was used. For CIFAR and SVHN datasets, ResNet (He et al., 2016) was used. For all datasets, we measured the training loss and the highest test accuracy during the training. We reported the mean and the standard deviation of the highest accuracies by repeating the training several times. Specifically, we repeated the training 10 times for MNIST dataset and 5 times for the other datasets. Table 1 summarizes the experiment results, and detailed explanations of the results for each dataset will be provided in the following sections.

For each gradient method, we selected the best initial learning rate using a grid search in a set of $\{0.1, 0.05, 0.01, 0.005, 0.001\}$ for all datasets. For the other hyperparameters of Adam, such as the exponential decay rate of Adam, we followed the settings of the original paper of Adam (Kingma & Ba, 2015). To set the additional hyperpameters of $L^4$-Adam and the bound functions of AdaBound,

Table 1: Test accuracy in percent of the neural networks trained by each gradient method. The highest test accuracy measured during the training is reported.

| Gradient method | MNIST | SVHN | CIFAR-10 | CIFAR-100 |
|---|---|---|---|---|
| RMSProp | $98.82 \pm 0.04$ | $94.51 \pm 0.14$ | $89.13 \pm 0.17$ | $64.61 \pm 0.09$ |
| Adam | $98.89 \pm 0.04$ | $94.96 \pm 0.13$ | $89.95 \pm 0.34$ | $67.35 \pm 0.18$ |
| $L^4$-Adam | $99.31 \pm 0.05$ | $95.68 \pm 0.09$ | $90.17 \pm 0.25$ | $66.79 \pm 0.81$ |
| AdaBound | $99.28 \pm 0.03$ | $95.48 \pm 0.12$ | $91.46 \pm 0.21$ | $69.71 \pm 0.16$ |
| SSO-SGD | $99.29 \pm 0.05$ | $\mathbf{96.41 \pm 0.08}$ | $\mathbf{94.43 \pm 0.15}$ | $\mathbf{74.96 \pm 0.36}$ |
| SSO-Adam | $\mathbf{99.33 \pm 0.04}$ | $95.75 \pm 0.06$ | $92.53 \pm 0.12$ | $70.82 \pm 0.41$ |

we used the recommended setting in their original papers (Rolinek & Martius, 2018; Luo et al., 2019). For MNIST dataset, the initial upper bound of SSO, $\epsilon^{(0)}$, was set to 0.5. For the other datasets, $\epsilon^{(0)}$ was fixed to 1. The decay factor in the upper bound decay, $\gamma$, was fixed to 0.95 for all datasets. The number of iterations for optimizing the step size in SSO was fixed to 20 ($I = 20$). We selected the best regularization coefficient ($\beta$) using a grid search for each gradient method on for all datasets. The batch size was fixed to 128 for all datasets. For MNIST dataset, we exploited a commonly used architecture of CNN with two convolution layer and one fully-connected output layer. For SVHN and CIFAR datasets, we used the ResNet with three residual blocks and one fully-connected output layer (ResNet-18). All experiments were conducted on NVIDIA GeForce RTX 2080 Ti. We used PyTorch to implement SSO and the author's source code for $L^4$-Adam[1] and AdaBound[2]. The source code of SSO and the experiment scripts are available at GitHubURL (*open after the review*).

## 5.1 DIGIT RECOGNITION

MNIST dataset is a widely used benchmark dataset for digit recognition. It contains 60,000 training instances and 10,000 test instances of $28 \times 28 \times 1$ size from 10 classes. As shown in Fig. 1-(a), $L^4$-Adam, AdaBound, SSO-SGD, and SSO-Adam rapidly reduced the training loss to zero on MNIST dataset. Although SSO-Adam showed the highest test accuracy, other gradient methods also showed similar accuracies because MNIST dataset is too simple and easy to fit the model (easy to achieve 99% test accuracy). For this reason, to accurately evaluate the effectiveness of each method, we compared the performance on SVHN dataset, which is also a widely used benchmark dataset for digit recognition and more realistic.

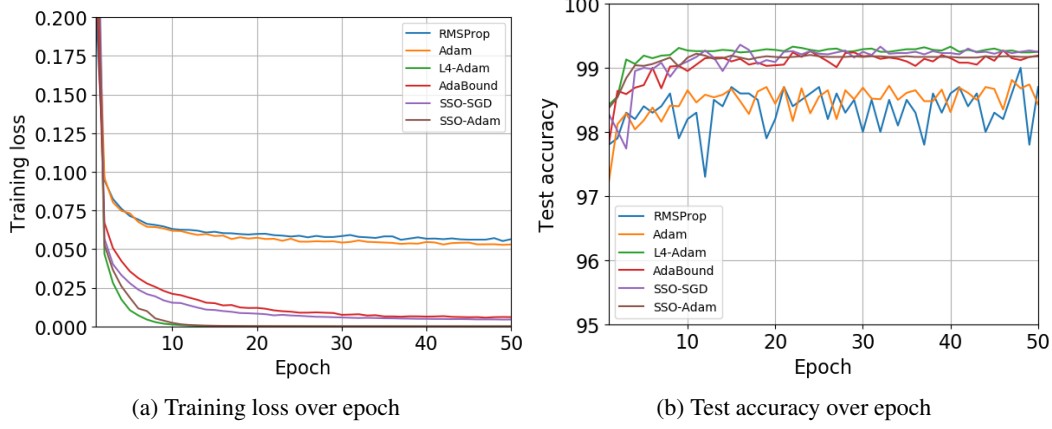

(a) Training loss over epoch      (b) Test accuracy over epoch

Figure 1: Training progress of CNNs on MNIST dataset.

---

[1] https://github.com/martius-lab/l4-optimizer
[2] https://github.com/Luolc/AdaBound

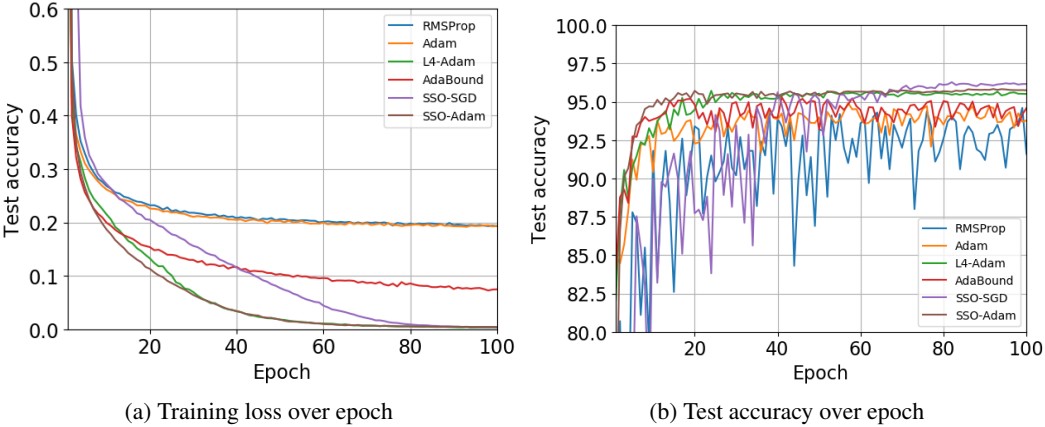

(a) Training loss over epoch

(b) Test accuracy over epoch

Figure 2: Training progress of ResNets on SVHN dataset.

SVHN dataset contains 73,257 training instances and 26,032 test instances of $32{\times}32{\times}3$ size from 10 classes. As shown in Fig. 2-(a), $L^4$-Adam, SSO-SGD, and SSO-Adam rapidly reduced the training loss to zero on SVHN dataset. Furthermore, both SSO-SGD and SSO-Adam outperformed all state-of-the-art competitors in the test accuracy. In the experiments, although $L^4$-Adam, SSO-SGD, and SSO-Adam all rapidly reduced the training loss, SSO-SGD and SSO-Adam showed better generalization performance than $L^4$-Adam.

## 5.2 OBJECT CLASSIFICATION

CIFAR datasets contain 50,000 training instances and 10,000 test instances of $32{\times}32{\times}3$ size. CIFAR-10 and CIFAR-100 contain 10 and 100 categories (classes), respectively. We also used ResNet with three residual blocks and one fully-connected layer for CIFAR datasets.

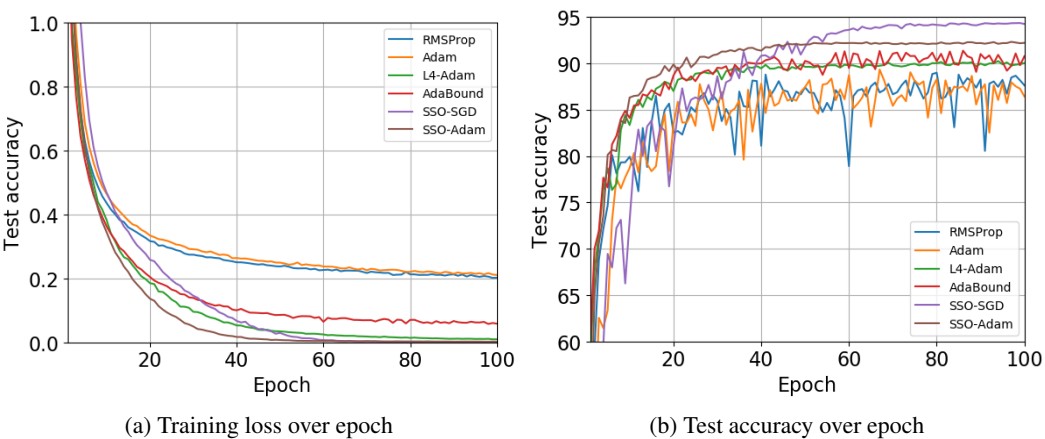

(a) Training loss over epoch

(b) Test accuracy over epoch

Figure 3: Training progress of ResNets on CIFAR-10 dataset.

On CIFAR-10 dataset, $L^4$, SSO-SGD, and SSO-Adam also rapidly reduced the training loss to zero (Fig. 3-a). Furthermore, SSO-SGD and SSO-Adam outperformed all competitors in the test accuracy (Fig. 3-a). Especially, SSO-SGD achieved about 3% improvement on the test accuracy compared to AdaBound that showed the highest test accuracy among the competitors.

As shown in Fig. 4-(b), both SSO-SGD and SSO-Adam outperformed all state-of-the-tart competitors in the test accuracy again. In particular, SSO-SGD achieved 5% improved test accuracy compared to $L^4$-Adam that showed the highest test accuracy among the competitors. In this experi-

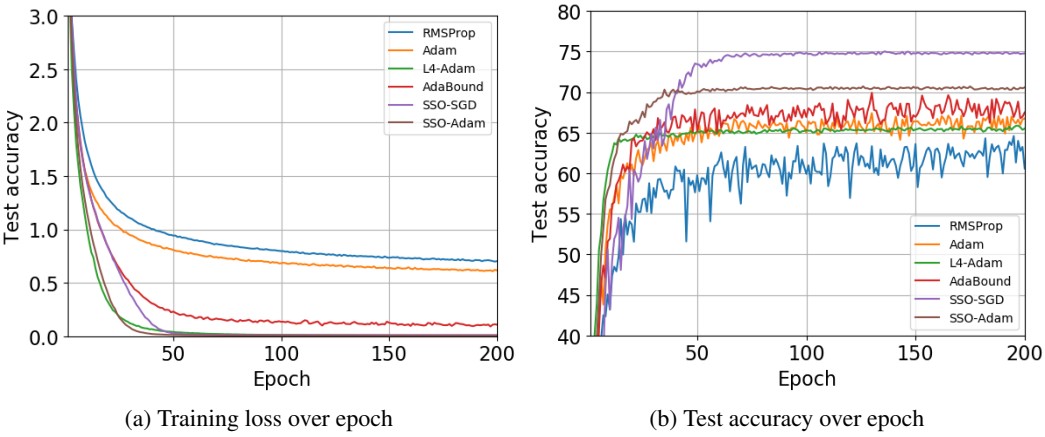

(a) Training loss over epoch  (b) Test accuracy over epoch

Figure 4: Training progress of ResNets on CIFAR-100 dataset.

ment, SSO-SGD and SSO-Adam showed better generalization performance than L$^4$-Adam because stochastic SSO is designed to the stochastic learning environments.

## 5.3 TRAINING PERFORMANCE OVER EXECUTION TIME

In this experiment, we measured the test accuracy over actual execution time to evaluate the usefulness of each method. Fig. 5 shows the results of the experiment on SVHN and CIFAR100 datasets.

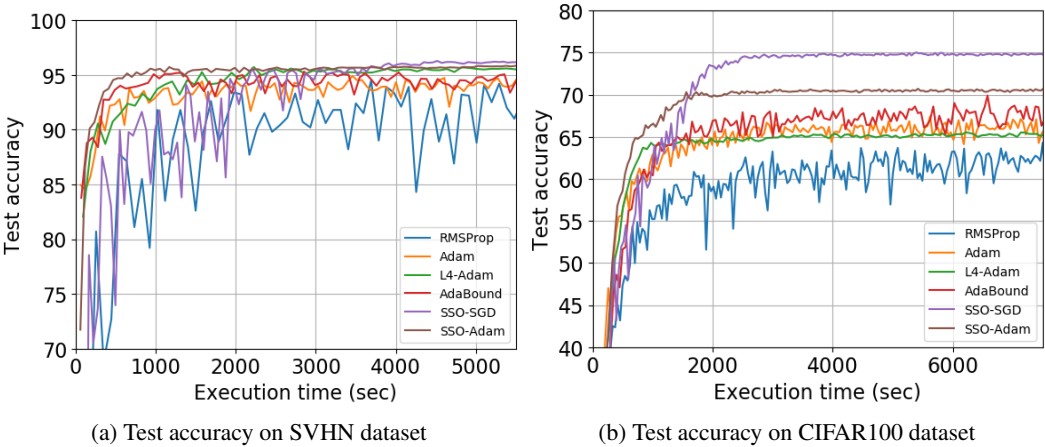

(a) Test accuracy on SVHN dataset  (b) Test accuracy on CIFAR100 dataset

Figure 5: Test accuracy over the execution time.

The experiment results are similar to the results in Fig. 2-(b) and 4. It shows that SSO is as efficient as existing step size adaptation methods. Furthermore, quantitatively, SSO-SGD and SSO-Adam required about 5,500 seconds execution time for 100 epochs like RMSProp and Adam, but L$^4$-Adam required about 9,000 seconds execution time that is 30% higher than the execution time of SSO-SGD and SSO-Adam. On CIFAR-100 dataset, SSO-SGD and SSO-Adam also required about 8,500 seconds execution time for 200 epochs like RMSProp and Adam, but L$^4$-Adam spent about 13,000 seconds.

## 5.4 INITIAL UPPER BOUNDS AND TRAINING PERFORMANCE

We checked the training performance on MNIST dataset by change the initial upper bound $\epsilon^{(0)}$ to measure the sensitivity of SSO for the hyperparameter. Fig. 6 shows the experiment results.

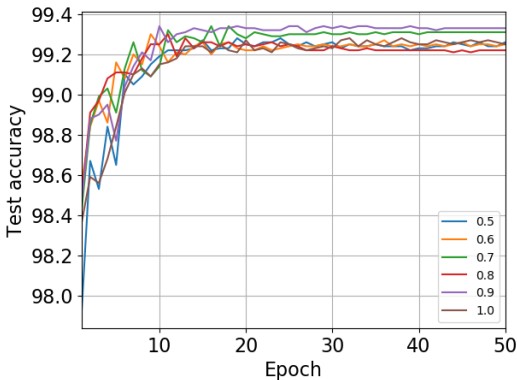

Figure 6: Changes on the test accuracy with the different initial upper bounds on MNIST dataset.

As shown in the result, SSO-Adam showed consistent test accuracy in 99.2%~99.35% for all initial upper bounds in $\{0.5, 0.6, 0.7, 0.8, 0.9, 1.0\}$.

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

# A   APPENDIX: OPTIMIZED STEP SIZE ON MNIST DATASET.

In this experiment, we measured the optimized step size of SSO rate for each epoch. We used SSO-Adam with the upper bound decay. Since the step size adaptation is executed by the number of mini-batches for each epoch, we presented maximum, mean, and minimum of the optimized step size for each epoch. Fig. 7 shows the optimized step sizes for each epoch. As shown in the result, the learning rate is strictly optimized within $[0, \epsilon]$ and gradually reduced over the epochs. Note that the current upper bounds are not shown because the maximums of the optimized step sizes overlap them in Fig. 7-(a).

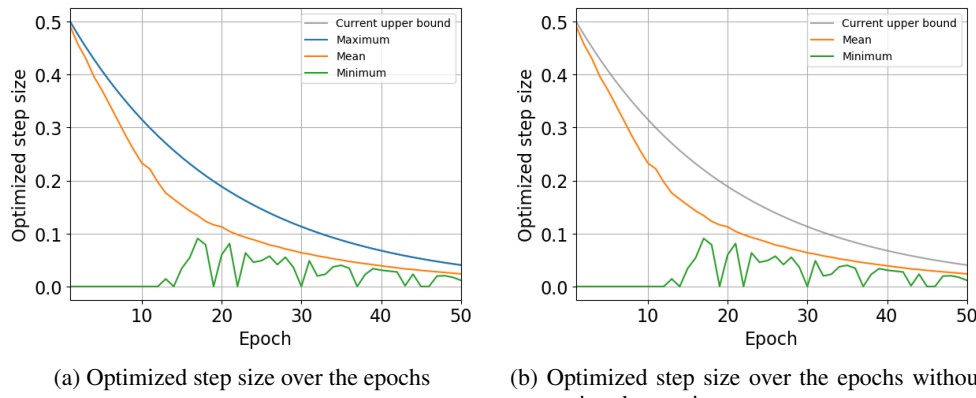

(a) Optimized step size over the epochs

(b) Optimized step size over the epochs without presenting the maximum

Figure 7: Maximum, mean, and minimum of the optimized step sizes for each epoch on MNIST dataset.

