# OpenReview forum: "Step Size Optimization"
_ICLR.cc/2020/Conference — Reject_

### Official Review · AnonReviewer2 · 2019-10-20
**Official Blind Review #2**

**Rating:** 3

**Review:**

First, I would like to point out that there has not been a conclusion or discussion section included, therefore the paper appears to be incomplete.
Aside from this the main contribution of the paper is a study on optimising the step size in gradient methods. They achieve this through the use of alternating direction method of multipliers. Given all the formulations provided, it appears as if this method does not rely on second order information or any probabilistic method.
An extension of the proposed method covers stochastic environments.
The results demonstrate some promising properties, including convergence and improvements on MNIST, SVHN, Cifar-10 and Cifar-100, albeit marginal improvements.
Although the results appear to be promising the overall structure of the paper and the method presented are based upon established techniques, therefore the technical contribution is rather limited.
I have read the rebuttal and answered to some of the concerns of the authors.

**Experience Assessment:**

I have read many papers in this area.

**Review Assessment: Checking Correctness Of Derivations And Theory:**

I assessed the sensibility of the derivations and theory.

**Review Assessment: Checking Correctness Of Experiments:**

I assessed the sensibility of the experiments.

**Review Assessment: Thoroughness In Paper Reading:**

I read the paper at least twice and used my best judgement in assessing the paper.

---

> ### Author Response · Authors · 2019-11-05
> **Response to review #2**
>
>
> We agree with your comment, so we will append a conclusion or discussion section.
>
> The performance improvement on CIFAR-10 and CIFAR-100 datasets is not marginal. Could you give a reference that achieves similar performance using RMSProp and Adam on ResNet-18?
>
> We first defined step size adaptation as a constrained optimization problem and converted it into a solvable problem by applying linearization and introducing slack variables. Then, we analyzed convergence of the proposed method with L2 regularization that is the most common regularization technique. To alleviate bad convergence problem, we developed the upper bound decay that is a generalized technique of the step size decay. Furthermore, we extended the proposed method into the stochastic learning environments. Thus, this paper is not just a list of existing methods. In this work, is there really no technical contribution? You should provide some references to criticize the performance improvement and technical contributions.

---

### Official Review · AnonReviewer4 · 2019-11-05
**Official Blind Review #4**

**Rating:** 3

**Review:**

This paper proposes a new step size adaptation in first-order gradient methods. The proposed method establishes a new optimization problem with the first-order expansion of loss function and the regularization, where the step size is treated as a variable.  ADMM is adopted to solve the optimization problem.

This paper should be rejected because (1) the proposed method does not show the convergence rate improvement of the gradient method with other step sizes adaptation methods. (2) the linearization of the objective function leads the step size to be small ($0<\eta<\epsilon$), which could slow down the convergence in some cases. (3) the experiments generally do not support a significant contribution. In table 1, the results of the competitor are not with the optimal step sizes. The limit grid search range could not verify the empirical superiority of the proposed method.


Minor comments:
The y-axis label of (a) panel in each figure is wrong. I guess it should be "Training loss ".

**Experience Assessment:**

I have published one or two papers in this area.

**Review Assessment: Checking Correctness Of Derivations And Theory:**

I carefully checked the derivations and theory.

**Review Assessment: Checking Correctness Of Experiments:**

I assessed the sensibility of the experiments.

**Review Assessment: Thoroughness In Paper Reading:**

I read the paper thoroughly.

---

> ### Author Response · Authors · 2019-11-05
> **Response to review #4**
>
>
> (1, 2) We don't know what you're pointing out. SSO always showed faster convergence speed than RMSProp and Adam. In addition, SSO consistently showed the performance improvement with relatively large initial learning rate (e.g., 0.5). Note that RMSProp and Adam commonly use very small initial learning rate (e.g., 0.001`). Thus, your comments are incorrect. Furthermore, SSO showed comparable convergence speed with L4-Adam and AdaBound while improving the generalization significantly.
>
> (3) We tuned hyperparameters of the competitors with the grid search and achieved the experimental results similar to other papers and GitHub repositories on CNN and ResNet-18. Especially, for L4-Adam and AdaBound, we used the best hyperparameters suggested in their original papers.
>
> You should provide clearer and more understandable review.

---

### Public Comment · ~Junxiang_Wang1 · 2019-10-12
**Interesting approach some points are confusing**

Dear author:
             Thank you for your interesting work. Step size optimization is an important topic. However, I find it difficult to understand some points in the paper.
1.  In page 2, why the dual variables lambda1 and lambda2 must be nonnegative? This may explain why there is a max operation in Equations 8 and 9.
2.  The convergence analysis of the proposed ADMM is confusing. As far as I know, Equation 4 is a multi-block ADMM (i.e., with more than two variables), and the multi-block ADMM is not guaranteed to converge. See the following paper for reference.
The direct extension of ADMM for multi-block convex minimization problems is not necessarily convergent
https://link.springer.com/article/10.1007/s10107-014-0826-5.
               Thanks.

---

### Public Comment · ~Jianlin_Su1 · 2019-10-23
**How do we explained why just expanding J(θ) but not Ω(θ)?**

An excellent job but still some confusions.

Why we just expanding J(θ − ηv) as  J(θ)−η g^T v, but not  Ω(θ − ηv) as  Ω(θ)−η g^T v?

I know you may want to get a closed form like eq.(14), but it is not a sufficient reason in my opinion. I think we must demonstrate that ignoring higher order of J(θ − ηv) is reasonable.

Meanwhile, can we do it while the loss has no regularizer term?

---

> ### Author Response · Authors · 2019-10-23
> **Linearization on the objective function**
>
>
> Linearization was not applied to obtain a closed-form solution, but to simplify the severely complex and nonlinear loss function. In this process, the approximation error inevitably occurs, so there is no reason to unnecessarily linearize the regularization term when it is simple (e.g., convex).
>
> SSO can be derived without regularization term, but we did not consider this training environment because most objective functions for training deep neural networks include the regularization term to improve training or testing performances.
>
> Thanks.

---

> > ### Public Comment · ~Jianlin_Su1 · 2019-10-24
> > **trivial if no regularization term**
> >
> > if without regularization term, the optimal η of eq.(3) is just ϵ, which is just a trivial result and makes no sense.

---

> > > ### Author Response · Authors · 2019-10-24
> > > **Unrealistic training environments**
> > >
> > >
> > > I agree with your concern because eta is trivial as epsilon for g^T v > 0 and zero for g^T v < 0 on the loss function without the regularization term. Nonetheless, this solution is mathematically correct due to the linearity of the simplified loss function.
> > >
> > > Furthermore, if the regularization term is added or minibatch is used, the solution is no longer trivial. Your concern is about the training environments that the training dataset perfectly represents the test dataset (non-overfitting) and the size of the training dataset is tiny (non-minibatch). These training environments are unrealistic in deep learning, so I did not consider them in designing the method.
> > >
> > > Thanks.

---

### Decision · Program_Chairs · 2019-12-19

**Decision:**

Reject

**Comment:**

The paper is rejected based on unanimous reviews.